# Cerebrospinal Fluid EV Concentration and Size Are Altered in Alzheimer’s Disease and Dementia with Lewy Bodies

**DOI:** 10.3390/cells11030462

**Published:** 2022-01-28

**Authors:** Antonio Longobardi, Roland Nicsanu, Sonia Bellini, Rosanna Squitti, Marcella Catania, Pietro Tiraboschi, Claudia Saraceno, Clarissa Ferrari, Roberta Zanardini, Giuliano Binetti, Giuseppe Di Fede, Luisa Benussi, Roberta Ghidoni

**Affiliations:** 1Molecular Markers Laboratory, IRCCS Istituto Centro San Giovanni di Dio Fatebenefratelli, 25125 Brescia, Italy; alongobardi@fatebenefratelli.eu (A.L.); rnicsanu@fatebenefratelli.eu (R.N.); sbellini@fatebenefratelli.eu (S.B.); rsquitti@fatebenefratelli.eu (R.S.); csaraceno@fatebenefratelli.eu (C.S.); rzanardini@fatebenefratelli.eu (R.Z.); lbenussi@fatebenefratelli.eu (L.B.); 2Neurology 5 and Neuropathology Unit, Fondazione IRCCS Istituto Neurologico Carlo Besta, 20133 Milan, Italy; marcella.catania@istituto-besta.it (M.C.); pietro.tiraboschi@istituto-besta.it (P.T.); giuseppe.difede@istituto-besta.it (G.D.F.); 3Service of Statistics, IRCCS Istituto Centro San Giovanni di Dio Fatebenefratelli, 25125 Brescia, Italy; cferrari@fatebenefratelli.eu; 4MAC Memory Clinic and Molecular Markers Laboratory, IRCCS Istituto Centro San Giovanni di Dio Fatebenefratelli, 25125 Brescia, Italy; gbinetti@fatebenefratelli.eu

**Keywords:** Alzheimer’s disease, dementia with Lewy bodies, frontotemporal dementia, extracellular vesicle, neurodegeneration, cystatin C, progranulin, nanoparticle tracking analysis, CSF, endo-lysosomal pathway

## Abstract

Alzheimer’s disease (AD), dementia with Lewy bodies (DLB) and frontotemporal dementia (FTD) represent the three major neurodegenerative dementias characterized by abnormal brain protein accumulation. In this study, we investigated extracellular vesicles (EVs) and neurotrophic factors in the cerebrospinal fluid (CSF) of 120 subjects: 36 with AD, 30 with DLB, 34 with FTD and 20 controls. Specifically, CSF EVs were analyzed by Nanoparticle Tracking Analysis and neurotrophic factors were measured with ELISA. We found higher EV concentration and lower EV size in AD and DLB groups compared to the controls. Classification tree analysis demonstrated EV size as the best parameter able to discriminate the patients from the controls (96.7% vs. 3.3%, respectively). The diagnostic performance of the EV concentration/size ratio resulted in a fair discrimination level with an area under the curve of 0.74. Moreover, the EV concentration/size ratio was associated with the p-Tau181/Aβ42 ratio in AD patients. In addition, we described altered levels of cystatin C and progranulin in the DLB and AD groups. We did not find any correlation between neurotrophic factors and EV parameters. In conclusion, the results of this study suggest a common involvement of the endosomal pathway in neurodegenerative dementias, giving important insight into the molecular mechanisms underlying these pathologies.

## 1. Introduction

Major neurodegenerative dementias are multifactorial conditions that share key underlying pathophysiological processes. A variety of triggers encompassing genetic, environmental, vascular, metabolic and inflammatory factors converge to activate common neurodegenerative mechanisms that take place in the brain of individuals affected by major neurodegenerative dementias and can partially explain their overlap [1]. Abnormal protein accumulation in the brain and inclusions that impair neuronal communication leading to cellular death constitute the main common neurodegenerative mechanisms [2,3].

Alzheimer’s disease (AD) represents the most common form of dementia in the elderly and is characterized by intra- and extra-cellular amyloid-β (Aβ) peptide aggregates forming the amyloid plaques and by phosphorylated tau protein accumulation in neurofibrillary tangles, pathognomonic of the disease [4,5]. These inclusions cause inflammatory and oxidative damage that are crucial for AD onset and progression [6]. Dementia with Lewy bodies (DLB) is one of the most common dementias after AD [7] and shares neuropathological characteristics with AD, such as amyloid plaques [8], but the main feature is the presence of α-synuclein inclusions in neurons, neurites, glia and presynaptic terminals. These inclusions cause the formation and the spreading of Lewy bodies widely in various brain areas [9]. Frontotemporal dementia (FTD), another major dementia, is typified by early-onset and by several protein inclusions such as tau, ubiquitin, Fused-in-Sarcoma (FUS) and TAR DNA-binding protein 43 (TDP-43) [10,11].

In recent years, extracellular vesicles (EVs) have been reported as a new concept in the biomarker field. Serving as transfer vehicles between cells of molecules, they represent a promising source of biomarkers for a number of diseases, including neurodegenerative disorders [12,13]. EVs consist of a heterogeneous family of small, cell-derived, membranous particles, including exosomes and microvesicles, the most studied subtypes of EVs [14]. Exosomes are carriers of misfolded neurotoxic proteins, such as Aβ, α-synuclein and tau proteins [15,16,17,18] and can, thus, be involved in the mechanisms underlying common pathophysiological processes at the basis of the major dementia overlap. Conversely, several studies have shown potential protective roles of EVs in neurodegenerative diseases by the removal of deleterious material derived from suffering tissues, or transporting neuroprotective/neurotrophic factors to distant regions, extending their effects and lifespan [19]. Cystatin C (CysC) is one of the neurotrophic factors associated with exosomes [20] exerting a protective role in response to neurotoxic conditions [21]. The co-localization of CysC and Aβ have been reported in both preclinical models and in brain amyloid plaques of AD patients, supporting the concept of the protective role of EV [22,23,24]. Similarly, progranulin (PGRN) is a neurotrophic factor associated with neurodegenerative diseases sustaining neuron survival, growth and anti-inflammatory processes [25,26]. Furthermore, in a recent study on human fibroblasts [27], altered levels of PGRN have been shown to cause a modification in EV intercellular communication supporting a possible disruption of the endo-lysosomal pathway. Brain-derived neurotrophic factor (BDNF) and glial-derived neurotrophic factor (GDNF) play an important role in the pathophysiology of neurodegenerative diseases, demonstrating a potential for therapeutic applications [28,29,30]: in a preclinical model of Parkinson’s disease, systemic administration of GDNF-expressing macrophages significantly improved the lifespan of mice [31]. In our previous study [32], we have provided evidence of a decrease in the concentration and an increase in the size of plasma EVs in AD, DLB and FTD, supporting the concept that an alteration in the intercellular communication mediated by EVs could represent a common molecular pathway underlying neurodegenerative dementias.

In this current study, we extended our investigation to the cerebrospinal fluid (CSF) compartment with the aim of investigating in more depth the role of EVs and neurotrophic factors in the three most common form of dementia.

## 2. Materials and Methods

### 2.1. Subjects

Human CSF samples from *n* = 36 AD, *n* = 30 DLB, *n* = 34 sporadic FTD patients and *n* = 20 samples from subjects with subjective memory complaints and a MMSE score >26, as control group (CTRL), were included in this retrospective study. All participants underwent CSF drawn by lumbar puncture and CSF samples were collected and stored at −80 °C following standard procedures. Patients were enrolled at the MAC Memory Clinic of the IRCCS Fatebenefratelli, Brescia, and at the Neurology 5/Neuropathology Unit, IRCCS Besta, Milan. Clinical diagnosis for probable AD, DLB and FTD was made according to international guidelines [33,34,35,36,37]. Participants provided written informed consent. The study protocol was approved by the local ethics committee (Prot. N. 111/2017).

### 2.2. CSF EV Isolation and Characterization

EV isolation was performed with the Total Exosome Isolation Kit (from other body fluids) (Invitrogen^TM^, Waltham, MA, USA) after optimization according to the manufacturer’s protocol. Briefly, 125 µL of CSF added with 75 µL of 0.2 µm filtered 1× phosphate-buffered saline (PBS) were centrifugated at 2000× *g* for 30 min at +4 °C, and subsequently 10,000× *g* at +4 °C for 30 min, and then transferred into new tubes, mixed with 200 µL of Exosome Precipitation Reagent (Invitrogen^TM^, Waltham, MA, USA) and incubated for 1 h at +2–8 °C. After incubation, samples were centrifugated at 10,000× *g* for 1 h at +4 °C. EV pellets were resuspended in 100 µL of 0.2 µm filtered 1× PBS and stored at +4 °C until nanoparticle tracking analysis (NTA) was performed. As the negative control, an aliquot of 100 µL of 1× PBS was processed as described above. For EV characterization, a representative CSF EV pellet was lysed with 30 µL of ice-cold Exosome Resuspension Buffer (Total Exosome RNA and Protein Isolation Kit, Invitrogen^TM^, Waltham, MA, USA) and stored at −20 °C. Alix and Calnexin expression were analyzed in EVs by Western blotting analysis according to standard protocols. Briefly, lysed EVs (40 µg) were separated using Bolt^TM^ 4–12% Bis-Tris Plus Gels (Invitrogen^TM^, Waltham, MA, USA) with MOPS SDS running buffer (Invitrogen^TM^, Waltham, MA, USA). Samples were electro-transferred onto nitrocellulose membranes (Thermo Fisher Scientific, Waltham, MA, USA) for 90 min at 90 V, the membranes were immunoblotted with primary antibodies overnight at +4 °C (anti-Alix, Abcam, Cambridge, UK) or for 2 h at +37 °C (anti-Calnexin, BD Biosciences, Franklin Lakes, NJ, USA) and then incubated with horseradish peroxidase-conjugated secondary antibodies (Invitrogen^TM^, Waltham, MA, USA) for 1 h at +37 °C. Immuno-positive bands were detected by ultra-sensitive enhanced chemiluminescence (Thermo Fisher Scientific, Waltham, MA, USA) according to the manufacturer’s instructions.

### 2.3. Nanoparticle Tracking Analysis (NTA)

EVs derived from CSF samples were analyzed with the NanoSight NS300 Instrument (Malvern, Worchestershire, UK). To gain an optimal reading range from 20–150 particles/frame, EV suspensions were diluted with 0.2 µm filtered 1X PBS. For each sample, 5 videos of 60 s were recorded, and the relative data were analyzed using NanoSight NTA Software 3.2 (Malvern, Worchestershire, UK). The instrument settings were optimized and kept constant between samples. Data collected consisted of particle concentration (particles/mL), average size (nm) and particle size distribution (D-values; D10, D50 and D90). Finally, raw concentration data (particles/mL) obtained from the instrument were normalized to calculate the EV concentration in CSF samples.

### 2.4. Biochemical Analyses

Cystatin C CSF concentration was measured using the Human Cystatin C Quantikine^®^ ELISA kit (R&D Systems^®^, Minneapolis, MN, USA) according to standard protocols; samples were diluted at 1:80. The mean intra-assay coefficient of variation (%CV) was <5% and the mean inter-assay CV% was <4%. BDNF and GDNF CSF concentrations were measured with Human Premixed Multiplex-Magnetic Luminex^®^ Assays (R&D Systems^®^, Minneapolis, MN, USA) following the manufacturer’s protocol; samples were diluted at 1:1. PGRN CSF concentration was measured with the Progranulin (human) ELISA kit (AdipoGen^®^, San Diego, CA, USA) following the manufacturer’s protocol; samples were diluted at 1:10. The mean intra-assay %CV was <3% and the mean inter-assay CV% was <5%. Aβ 40, Aβ 42, p-Tau 181 and Tau CSF concentrations were measured using Innotest ELISA kits (Fujirebio, Tokyo, Japan) following the manufacturer’s protocol. All samples were analyzed in duplicate.

### 2.5. Statistical Analysis

Normality assumption of continuous variables was evaluated with graphical inspection and the Kolmogorov–Smirnov test. The linear model or generalized linear model (for normal or non-normal distributed variable, respectively) adjusted for age were used for the comparison across the four subject groups. Bonferroni post-hoc tests were applied. The chi-square test was used to assess the association between the demographic characteristics (categorical variables) of the subjects within the four groups. A classification tree (CT) [38] was applied to detect the best (in terms of classification performance) predictors for discriminating the controls versus patients group. The CT method was carried out on the diagnostic group as a categorical dependent variable depending on categorical and/or quantitative covariates. The output of the CT was given by different classification pathways (defined by estimated covariate cut-offs), and for each of them the probability of the most likely diagnostic group was provided. In addition, diagnostic performance of EV concentration and EV size in discriminating across the groups was assessed by area under the curve (AUC) obtained by receiver operating characteristic (ROC). Finally, partial correlation (age controlled) analyses were performed on the EV concentration/size ratio, neurotrophic factors and CSF core biomarkers for AD. All analyses were performed by SPSS software and significance set at 0.05.

## 3. Results

### 3.1. CSF EV Size and Concentration Are Altered in Dementia Patients

Clinical and demographic variables of the participants under study are shown in Table 1. Groups did not differ for sex but differed for age, with the FTD patients being younger than the other groups. Isolated CSF EVs were Alix+ (a cytosolic protein recovered in EVs) and Calnexin- (an endoplasmic reticulum residential protein, absent in EVs) (Appendix A). CSF EV concentration was increased in patients (PTS) compared to CTRL (*p* = 0.002) despite not reaching the significance level in the FTD group (Figure 1a) (*p* = 0.001, CTRL vs. AD, DLB, *p* < 0.01). EV size had an opposite trend, being lower in PTS than in CTRL (*p* < 0.001) even though not significant in the FTD group (Figure 1b) (*p* = 0.021, CTRL vs. AD, DLB, *p* < 0.05). EV concentration and size distribution are depicted in the Appendix A. A Spearman’s correlation test confirmed a negative association between EV concentration and EV size (*p* < 0.001, r = −0.39). EV concentration/size ratio (Figure 1c) was increased in PTS compared to CTRL (*p* < 0.001), specifically in AD and DLB (*p* < 0.001, CTRL vs. AD, DLB, *p* < 0.01).

### 3.2. Cystatin C and Progranulin CSF Are Altered in DLB and AD

Cystatin C concentration was decreased in DLB samples compared to all other groups (Figure 2a) (*p* < 0.001, DLB vs. AD, FTD, *p* < 0.001; DLB vs. CTRL, *p* < 0.05). Progranulin concentration was increased in AD samples compared to DLB samples (Figure 2b) (*p* = 0.006, AD vs. DLB, *p* < 0.01). BDNF and GDNF were not detectable in any CSF samples.

### 3.3. EV Parameters Are Able to Discern Patients from Controls

In order to evaluate the capacity of demographic, EV and neurotrophic factors variables to classify subjects into patients or controls, different CTs were performed, including the group of patients separately (AD, DLB and FTD) or collapsed in PTS. The best CT (in terms of smaller classification error) was obtained with PTS and CTRL groups revealing EV parameters as the best predictors: EV size smaller than 114.9 nm was able to classify the patients from the controls (96.7% vs. 3.3%) (Figure 3). To estimate the diagnostic performance of the EV concentration/size ratio to discriminate PTS from CTRL we performed ROC analyses: considering the whole patient group, the AUC was 0.74, with a 75.0% specificity and a 69.0% sensitivity with a cut-off point of 2.90 × 10^6^. Considering each diagnostic group, we calculated an AUC of 0.73 for AD, 0.82 for DLB and 0.68 for FTD (Appendix A).

### 3.4. EV Parameters Are Related to CSF Core Biomarkers for AD

To evaluate the interaction of the biological variables under study and the diagnostic groups, partial correlation analyses were performed among the CSF EV concentration/size ratio, CysC and PGRN (neurotrophic factor), Aβ 40, Aβ 42, p-Tau 181, Tau, Aβ 42/Aβ 40 and p-Tau 181/Aβ 42 (core biomarkers for AD). The analyses were carried out for the entire study group as well as in PTS, in each diagnostic group (AD, DLB, FTD) and in CTRL. A positive correlation was found in PTS between EV concentration/size and p-Tau 181/Aβ 42 ratios (age adjusted; r = 0.230, *p* = 0.031). In the stratified analysis, the AD group resulted in being the only group with EV concentration/size and p-Tau 181/Aβ 42 ratios significantly correlated (age adjusted, AD: r = 0.358, *p* = 0.035). CysC and PGRN were not correlated with EV parameters.

## 4. Discussion

Emerging data argue for an interdependence between the production of EVs and the endosomal pathway in the brain [19]. We recently reported that genes controlling key endo-lysosomal processes (i.e., protein sorting/transport, clathrin-coated vesicles uncoating, lysosomal enzymatic activity regulation) might be involved in AD, FTLD and DLB pathogenesis, thus, suggesting an etiological link behind these diseases [39]. In this study, we demonstrated alterations of EVs and neurotrophic factors in CSF of patients with neurodegenerative dementias. We analyzed CSF EVs in patients affected by AD, DLB and FTD, and we found an altered EV profile in the patients’ CSF. Specifically, EV concentrations were higher in AD and DLB while CSF EV size was lower in AD and DLB compared to the controls. EV size was the EV variable with the best capacity to discriminate the patients affected by dementia from the controls: in CT analysis, the EV size resulted in the most predictive variable able to classify the patients from the controls (96.7% vs. 3.3%, respectively). The EV concentration/size ratio could discriminate the patients affected by dementia from the controls, reaching a fair discrimination level (AUC 0.74), with a 75.0% specificity and a 69.0% sensitivity with a cut-off point of 2.90 × 10^6^. According to the Working Group on “Molecular and Biochemical Markers of AD”, in order to be clinically useful, a diagnostic marker should have sensitivity and specificity approaching or exceeding 80–85% [40]. Thus, the specificity and sensitivity of the CSF EV concentration/size ratio presented in this study resulted in being suboptimal.

We previously described plasma EV alterations in neurodegenerative dementias with a significant reduction in EV concentration and larger EVs in AD, DLB and FTD patients [32]; herein, we observed an inverse alteration of EV variables in CSF with a higher EV concentration and smaller EV size in patients. Furthermore, in plasma, EV concentration was 3 orders of magnitude higher than in CSF, and the diagnostic performance of the EV concentration/size ratio was higher (AUC 0.86) with a sensitivity of 83.3% and a specificity of 86.7%. EV transfer from the peripheral circulatory system to the central nervous system is rare under physiological conditions; however, inflammatory processes may compromise the blood–brain barrier (BBB) allowing EV transport from the periphery to the brain: for example, EVs derived from erythrocytes can cross the BBB, contain a large amount of α-synuclein and may contribute to Parkinson pathology [41]. Thus, the observed EV increase in the CSF could be due to an altered brain–periphery communication and reflect biological processes of neurodegeneration occurring in the brain.

Current results of the association of the EV concentration/size ratio with the p-Tau 181/Aβ 42 ratio, as CSF biomarkers of AD [42], are in this direction. The EV concentration/size ratio with the p-Tau 181/Aβ 42 ratio correlation was found for the entire PTS group but the stratified analysis for dementia diagnostic categories revealed that this effect was driven by the AD group. In general, our results show a slight difference in EV variables in FTD with respect to AD and DLB.

In line with our results, EVs, and specifically microvesicles released by reactive microglia, were demonstrated to be increased in subjects with mild cognitive impairment (MCI) and in AD, compared to the controls. Moreover, EVs were associated with CSF biomarkers of AD; specifically, a negative correlation between EV concentration and CSF Aβ1-42 levels was found in the MCI group, while CSF Tau levels (t-Tau and p-Tau) were positively correlated with EV concentration both in MCI and AD [43,44]. Moreover, CSF EV were associated with brain atrophy and white matter tract damage, thus, suggesting that the release of EV by microglia might participate in AD neurodegeneration. A number of studies have shown that EV secretion of aggregation-prone proteins such as Aβ, α-synuclein, Tau or prion protein take places in neurodegenerative dementias [45,46,47,48]. Aside from the common mechanisms of aggregation-prone proteins spreading within the brain, EVs have been proposed to contribute to trans-synaptic Tau transmission and the propagation of Tau pathology in AD, from the entorhinal cortex to the hippocampus and the surrounding areas [49]. The fascinating hypothesis that EVs may constitute a prion-like mechanism for the spreading of disease proteins is indeed counterbalanced by evidence indicating that EV, and specifically exosomes, may act as scavengers of neurotoxic soluble Aβ [50]. Thus, whether exosomes and EV increase or decrease the detrimental action of Aβ is still a matter of debate [51].

We then investigated whether an alteration of neurotrophic factors in CSF might also be detected in AD, DLB and FTD, and if their levels could be related to EV release. We observed a decrease in CysC in dementia patients and more specifically in DLB patients; in line with our results, lower CysC CSF levels have been previously shown to be associated with DLB [8,52]. Regarding PGRN, AD patients showed an increased value of PGRN compared to the DLB group; AD had the highest levels of PGRN despite not reaching a statistical threshold compared to the other groups. The absence of a statistical significance between groups may be due to the small number of cases used in the analysis, representing a limitation in this study. Of note, PGRN CSF concentration has been shown to be increased with microglia activation in AD [53,54] and in the progression of the disease [55]. In the present study, we did not find any correlation between neurotrophic factors and EV concentration and size. In contrast, in plasma, we described a positive correlation of CysC with EV release in patients [32], suggesting that this neuroprotective factor, as well as an anti-amyloidogenic protein, might affect EV release. In line with this observation, it has been previously demonstrated in mice models that CysC enhances brain EV secretion, resulting in a protective effect [21]. The present study, investigating the correlation of EVs and CysC in the CSF of patients, did not confirm this paradigm.

In conclusion, we confirmed a common involvement of the endosomal pathway in neurodegenerative dementias, as suggested by the alterations of CSF EVs in AD and DLB. However, the role of the EVs has to be understood in more depth since the literature is still contradictory about their protective/pathogenic function in neurodegeneration. Since we described blood EVs (EV concentration/size) as a cross-disease biomarker with high diagnostic performance, more studies are needed in order to clarify the molecular mechanism underlying the observed effect at the peripheral level and the relationship with the inverse alteration observed at central level in CSF. Thus, although CSF EV variables show a fair diagnostic performance, plasma EVs could represent a better biomarker due the more feasible access for sampling and the better diagnostic accuracy. However, the results of this study indicate CSF EVs as a promising source for further investigation into the interaction of EVs and aggregation-prone proteins to give insight into the molecular mechanisms underlying the pathology.

## Figures and Tables

**Figure 1 cells-11-00462-f001:**
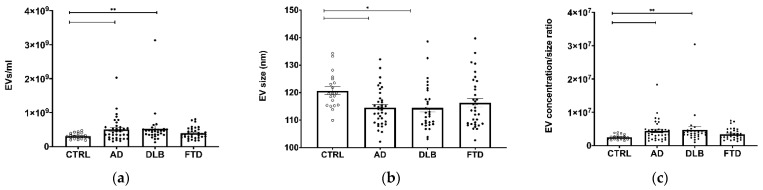
EV size and concentration are altered in AD and DLB CSF. (**a**) NTA analysis of EV concentration in CSF samples. A statistically significant increase in EV concentration was observed in AD and DLB groups compared to CTRL group. (**b**) EV size was significantly decreased in AD and DLB compared to CTRL group. (**c**) EV concentration/size ratio was increased in AD and DLB compared to CTRL group. Average ± SEM; * *p* < 0.05, ** *p* < 0.01. Bar plots represent raw data while the post-hoc *p*-values were obtained by generalized linear model adjusted for age.

**Figure 2 cells-11-00462-f002:**
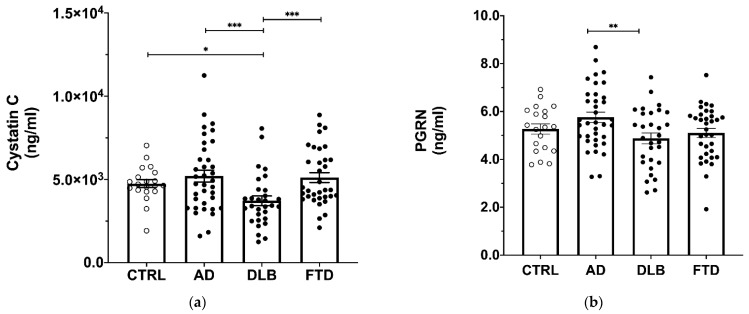
Neurotrophic factors levels in CSF. (**a**) Measurement of Cystatin C in CSF. A statistically significant decrease was observed in DLB compared to all other groups. (**b**) PGRN levels in CSF. PGRN was significantly increased in AD compared to DLB group. Average ± SEM; * *p* < 0.05, ** *p* < 0.01, *** *p* < 0.001. Bar plots represent raw data while the post-hoc *p*-values were obtained by generalized linear model adjusted for age.

**Figure 3 cells-11-00462-f003:**
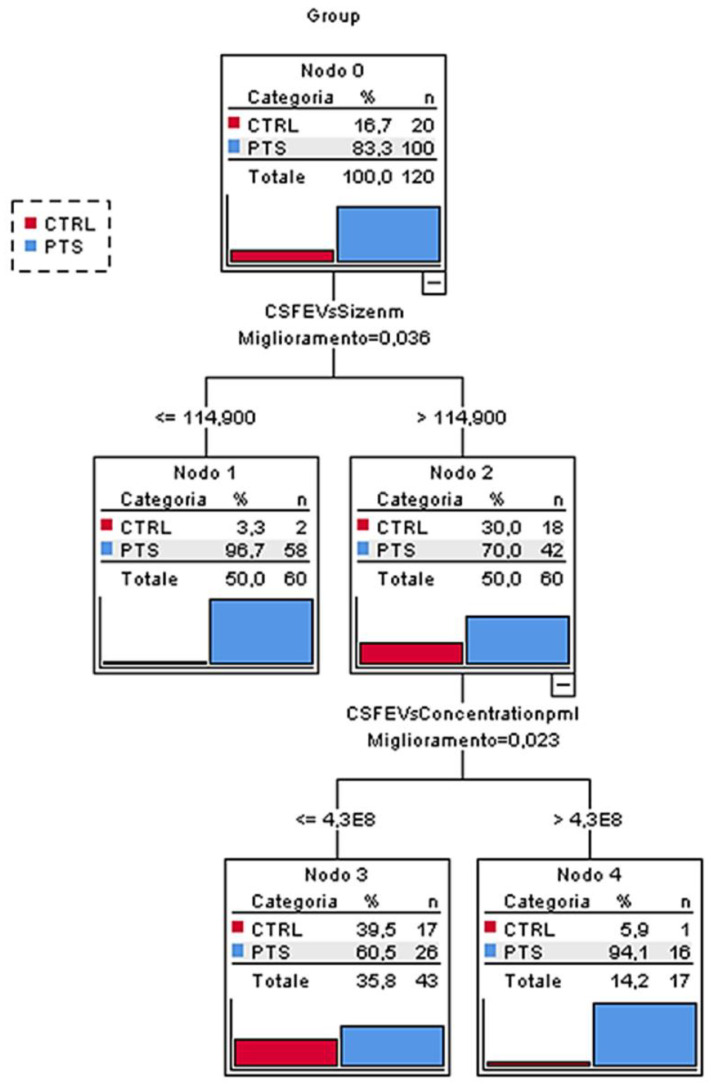
Classification tree. EV size resulted in being the most predictive variable among all the ones significantly associated to the groups. CTRL, controls; PTS, patients; CSFEVs size (nm), EV size (nm); CSFEVs concentration (p/mL), EV concentration (EVs/mL).

**Table 1 cells-11-00462-t001:** Clinical, demographic, and biological variables of patients and controls.

	CTRL (*n* = 20)	AD (*n* = 36)	DLB (*n* = 30)	FTD (*n* = 34)	*p*-Value
Sex (M:F) ^£^	9:11	17:19	15:15	15:19	0.969
Age, years ^$^	69.1 ± 8.7	70.4 ± 9.3	73.7 ± 5.6	65.2 ± 8.2	<0.001
Disease onset, years ^$^	-	66.5 ± 8.9	70.8 ± 7.1	62.3 ± 8.0	0.003
Education, years ^$^	7.7 ± 3.6	6.8 ± 3.5	8.0 ± 3.9	9.0 ± 5.0	0.210
MMSE ^$^	28.1 ± 1.7	19.0 ± 5.3	22.5 ± 6.3	18.5 ± 7.1	<0.001
Aβ 42, pg/mL ^$^	553.61 ± 207.81	366.75 ± 138.86	463.48 ± 229.50	498.22 ± 265.40	0.007 ^#^
Aβ 40, pg/mL ^$^	2314.64 ± 1366.98	3048.80 ± 1427.30	2494.66 ± 1321.17	1254.12 ± 513.79	<0.001 ^#^
p-Tau 181, pg/mL ^$ %^	48.80 ± 16.5	82.76 ± 32.90	52.40 ± 16.23	65.32 ± 38.83	<0.001 ^#^
Tau, pg/mL ^$ %^	258.98 ± 152.13	531.23 ± 205.97	306.57 ± 155.88	457.01 ± 318.16	<0.001 ^#^
Aβ 42/Aβ 40 ratio ^$^	0.33 ± 0.13	0.10 ± 0.04	0.23 ± 0.10	0.33 ± 0.16	<0.001 ^#^
p-Tau 181/Aβ 42 ratio ^$ %^	0.11 ± 0.08	0.27 ± 0.21	0.15 ± 0.09	0.19 ± 0.17	<0.001 ^#^
EV Concentration, EVs/mL ^$^	3.02 × 10^8^ ± 8.43 × 10^7^	5.02 × 10^8^ ± 3.50 × 10^8^	5.27 × 10^8^ ± 5.16 × 10^8^	3.96 × 10^8^ ± 1.66 × 10^8^	0.001 ^#^
EV Size, nm ^$^	120.67 ± 6.32	114.55 ± 6.67	114.45 ± 8.20	116.28 ± 9.06	0.021 ^#^
EV concentration/size ratio ^$^	2.52 × 10^6^ ± 7.64 × 10^5^	4.43 × 10^6^ ± 3.17 × 10^6^	4.74 × 10^6^ ± 5.06 × 10^6^	3.47 × 10^6^ ± 1.59 × 10^6^	<0.001 ^#^
Cystatin C, ng/mL ^$ %^	4749.02 ± 1078.63	5203.86 ± 2138.29	3723.72 ± 1576.36	5113.22 ± 1729.21	<0.001 ^#^
PGRN, ng/mL ^& %^	5.27 ± 0.95	5.76 ± 1.29	4.87 ± 1.25	5.10 ± 1.10	0.006 ^#^

CTRL, controls; AD, Alzheimer’s disease patients; DLB, dementia with Lewy bodies patients; FTD, frontotemporal dementia patients; MMSE, Mini-Mental State Examination score. ^£^ Chi-squared test; ^#^ Model with Age as covariate; ^&^ Modelled with linear model; ^$^ Modelled with generalized linear model; ^%^ Age is significant for this model. Means ± standard deviation.

## Data Availability

All study data, including raw and analyzed data, will be available upon reasonable request.

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
