# Peer review of "Cerebrospinal Fluid EV Concentration and Size Are Altered in Alzheimer’s Disease and Dementia with Lewy Bodies"

_cells, 2022, doi:10.3390/cells11030462_

Round 1

Reviewer 1 Report

Ref. Cells-1517170

The role of extracellular vesicles (EV) is currently a hot topic in neurodegenerative disorders, since it is related to many different mechanisms including proteinopathy-related toxicity, microglial activation, neurotrophic factors and prion-like spreading of misfolded proteins. The authors investigated cerebrospinal fluid EV in Alzheimer’s disease, dementia with Lewy bodies and frontotemporal dementia vs controls. They found that EV size is the best parameter for the discrimination between patients and controls, while the EV concertation/size ratio showed a significant positive correlation with the p-tau/Aβ42 ratio (a marker of AD) in AD patients.

This paper is very interesting, providing further evidence for the involvement of the endosomal pathway in neurodegeneration. The study is well designed and conducted and the statistics excellent.

Minor comments:

(a) The number of patients with AD provided in the Abstract (30) is not in agreement with the number reported in the text (typo error). Please correct or remove the number 30.

(b) I could not find in the text how the classical CSF biomarkers (tau, p-tau, Aβ42, Αβ40) were measured. Please provide this information

(c) It would be preferable to show the ROC curve(s).

(d) According to the Working Group on “Molecular and Biochemical Markers of AD”, in order to be clinically useful, a diagnostic marker should have sensitivity and specificity approaching or exceeding 80-85%. (The Ronald and Nancy Reagan Research Institute of the Alzheimer’s Association and the National Institute of Aging Working Group. Consensus report of the Working Group on: “Molecular and Biochemical Markers of Alzheimer’s Disease”. Neurobiol Aging. 1998;19:109–116). The sensitivity and specificity observed in the present study may be suboptimal, although still work has to be done for the evaluation of the diagnostic value. Please add a short comment in the Discussion section.

Reviewer 2 Report

In the presented study Longobardi and colleagues analyzed extracellular vesicles (EVs) alongside several biomarkers (Aß, tau, Cystatin C and Progranulin) in different types of dementia (AD, DLB and FTD) in comparison to controls. They conclude that EVs from CSF and their size and concentration is a suitable biomarker to differentiate between AD, DLB (and partly FTD) and controls.

This study in principle is well conducted however, I have some major concerns that need to be addressed before publication:

  1. line 51: The authors state that DLB is the second most common dementia behind AD. I am not sure whether this is true for all regions and DLB is rather as common as vascular, secondary and mixed dementias. Therefore, I would recommend rephrasing the sentence to "DLB is one of the most common dementias after Alzheimer's disease"
  2. Isolation of EVs:
    1. According to the protocol, no size exclusion chromatography was performed. Can the authors please state why?
    2. How can the authors be sure that they only analyze EVs in the Nanosight and not other particles? Did they perform analysis of EV specific proteins like Alix, Flotilin-1 or CD9 (or others)?
    3. Did the authors check for isolation efficacy by measuring the protein concentration?
  3. How was the performance of ELISAs concerning intra- and interassay variability?
  4. Please check the resolution of figure 3
  5. The ROC analysis of EV parameters yields an AUC of 0.74 with a sensitivity of 69% and specificity of 75%. This is okay but not really helpful as a diagnostic biomarker. Can the authors please compare this to the AUC of Aß40, 42 and the ratio and then discuss the utility of EV parameters as diagnostic biomarker more critically.
  6. In the discussion, the authors state that higher EV concentration in the CSF of dementia patients might be due to altered brain-periphery communication. In the next paragraph, they also discuss the potential influence of microglial activation on the EV concentration. Can the authors please discuss more in detail which processes by EVs reflect CNS processes? It might be interesting to analyze a microglial marker (e.g. sTREM2) and correlate it to EV parameters to check for the influence of microglial activation on EVs.
  7. The authors conclude that plasma EVs are more promising as diagnostic biomarkers than CSF EVs due to their accessibility and better performance. However, the authors should discuss the limited pathophysiological insights plasma EVs give and also compare their performance to other serum biomarkers in dementia (e.g. NfL and GFAP).

In general, the authors need to (if possible) better describe the advantage of EV analysis in dementia. I do not see the suitability of EV parameters as a diagnostic biomarker as their discrimination between dementia patients and control is rather poor and worse than existing biomarkers. If they offer insights into the pathophysiology of dementia this needs to discuss in more detail and how specific quite general parameters like size and concentration of EVs are.

Round 2

Reviewer 2 Report

I thank the authors for their detailed answers.

However, I still think that it is not sufficient to refer to other publications or information given by the manufacturer to control for the isolation efficacy of EVs. Control experiments (Western Blot of EV specific proteins or at least determination of protein concentrations) need to be included in this study before it can be accepted for publication.
